# Evaluating the Diagnostic Value of Lymphocyte Subsets in Bronchoalveolar Lavage Fluid and Peripheral Blood Across Various Diffuse Interstitial Lung Disease Subtypes

**DOI:** 10.3390/biom15010122

**Published:** 2025-01-14

**Authors:** Sonoko Harada, Motoyasu Kato, Kazuyuki Nakagome, Hitoshi Sasano, Yuki Tanabe, Tomohito Takeshige, Yuuki Sandhu, Kei Matsuno, Shoko Ueda, Sumiko Abe, Takayasu Nishimaki, Shun Shinomiya, Jun Ito, Sachiko Miyake, Ko Okumura, Makoto Nagata, Kazuhisa Takahashi, Norihiro Harada

**Affiliations:** 1Department of Respiratory Medicine, Juntendo University Faculty of Medicine and Graduate School of Medicine, Tokyo 113-8421, Japan; snharada@juntendo.ac.jp (S.H.); mtkatou@juntendo.ac.jp (M.K.); h-sasano@juntendo.ac.jp (H.S.); yutanabe@juntendo.ac.jp (Y.T.); takeshig@juntendo.ac.jp (T.T.); ysando@juntendo.ac.jp (Y.S.); kmatsuno@juntendo.ac.jp (K.M.); sk-ueda@juntendo.ac.jp (S.U.); su-abe@juntendo.ac.jp (S.A.); t-nishimaki@juntendo.ac.jp (T.N.); moisture@juntendo.ac.jp (J.I.); kztakaha@juntendo.ac.jp (K.T.); 2Atopy (Allergy) Research Center, Juntendo University Faculty of Medicine and Graduate School of Medicine, Tokyo 113-8421, Japan; kokumura@juntendo.ac.jp; 3Research Institute for Diseases of Old Ages, Juntendo University Faculty of Medicine and Graduate School of Medicine, Tokyo 113-8421, Japan; 4Department of Respiratory Medicine, Saitama Medical University, Saitama 350-0451, Japan; nakagomek-tky@umin.ac.jp (K.N.); shinomiso4038@gmail.com (S.S.); favre4mn@saitama-med.ac.jp (M.N.); 5Department of Immunology, Juntendo University Faculty of Medicine and Graduate School of Medicine, Tokyo 113-8421, Japan; s-miyake@juntendo.ac.jp

**Keywords:** diffuse interstitial lung diseases, chronic eosinophilic pneumonia, sarcoidosis, idiopathic interstitial pneumonia, chronic hypersensitivity pneumonitis, connective tissue disease-associated interstitial pneumonia

## Abstract

Diffuse interstitial lung diseases (ILD) include conditions with identifiable causes such as chronic eosinophilic pneumonia (CEP), sarcoidosis (SAR), chronic hypersensitivity pneumonitis (CHP), and connective tissue disease-associated interstitial pneumonia (CTD), as well as idiopathic interstitial pneumonia (IIP) of unknown origin. In non-IIP diffuse lung diseases, bronchoalveolar lavage (BAL) fluid appearance is diagnostic. This study examines lymphocyte subsets in BAL fluid and peripheral blood of 56 patients with diffuse ILD, excluding idiopathic pulmonary fibrosis (IPF), who underwent BAL for diagnostic purposes. Patients were classified into CEP, SAR, CHP, CTD, and IIP groups, and clinical data, BAL cell analysis, and peripheral blood mononuclear cell analysis were compared. Eosinophils and type 3 innate lymphocytes (ILC3s) were significantly increased in the BAL fluid of the CEP group. Receiver operating characteristic curve analysis identified eosinophils ≥ 8% in BAL cells and ILC3s ≥ 0.0176% in the BAL lymphocyte fraction as thresholds distinguishing CEP. SAR patients exhibited significantly elevated CD4/CD8 ratios in the BAL fluid, with a ratio of 3.95 or higher and type 1 innate lymphoid cell frequency ≥ 0.254% as differentiation markers. High Th1 cell frequency (≥17.4%) in BAL lymphocytes in IIP, elevated serum KL-6 (≥2081 U/mL) and SP-D (≥261 ng/mL) in CHP, and increased BAL neutrophils (≥2.0%) or a low CD4/CD8 ratio (≤1.2) in CTD serve as distinguishing markers for each ILD. Excluding CEP and SAR, CD4^+^ T cell frequencies, including Th1, Th17, and Treg cells in peripheral blood, may differentiate IIP, CHP, and CTD.

## 1. Introduction

Interstitial lung diseases (ILDs) comprise a heterogeneous group of disorders affecting the lung parenchyma, characterized by varying patterns of inflammation and fibrosis which can result in progressive lung scarring [1,2,3,4,5]. When the etiology is unknown, these diseases are termed idiopathic, including patients with idiopathic pulmonary fibrosis (IPF), idiopathic nonspecific interstitial pneumonia (NSIP), cryptogenic organizing pneumonia (COP), and so on [6]. Sarcoidosis (SAR) is a chronic multiorgan disease characterized by non-necrotizing granulomatous inflammation in various organs, and it has diverse clinical manifestations. Chronic eosinophilic pneumonia (CEP) is characterized by chronic and abnormal accumulation of eosinophils in the lung parenchyma and interstitium. Both diseases are considered ILDs; however, their causes remain unknown. Conversely, some ILDs are associated with identifiable causes, such as connective tissue disease-related ILD (CTD) and chronic hypersensitivity pneumonitis (CHP) (from 2020 onward, the condition has been categorized as fibrotic hypersensitivity pneumonitis) [7,8]. Among 17 methodologically diverse studies on ILD incidence, prevalence, and subtype frequencies, ILD incidence ranged from 1 to 31.5 per 100,000 person-years, and prevalence from 6.3 to 71 per 100,000 people [9]. IPF and SAR were most prevalent in North America and Europe, while hypersensitivity pneumonitis was more frequent in Asia [9]. CTD showed the greatest geographic variability, ranging from 7.5% in Belgium to 33.3% in Canada and 34.8% in Saudi Arabia [9].

High-resolution computed tomography (HRCT) of the chest can reveal the diagnostic features of ILD; however, bronchoalveolar lavage (BAL) and biopsies are essential for a precise diagnosis in many cases [10]. BAL, conducted via bronchoscopy, is a minimally invasive and well-tolerated procedure for assessing diffuse infiltrative lung diseases [4,5,11,12,13]. BAL is a valuable tool for the differential diagnosis of ILDs, including pulmonary alveolar proteinosis and diffuse alveolar hemorrhage, as it detects disease-specific factors [11]. Furthermore, analyzing BAL cell patterns and lymphocyte phenotyping often offers valuable insights for differentiating among various ILDs [11]. In particular, BAL cellularity and lymphocyte immunophenotyping provide crucial insights into the inflammatory status of the lungs [4,5,14,15,16]. Cytological analyses of BAL samples reveal the cellular composition within the alveoli, with lymphocytes being the most accessible cells in the bronchoalveolar space. These lymphocytes represent approximately 5% of the total circulating lymphocyte pool in humans [4,5,17,18]. For instance, in SAR, CD4^+^ T helper lymphocyte accumulation in tissues and an elevated CD4/CD8 ratio in BAL support the diagnosis [19,20]. Lymphocytosis in the BAL fluid may aid in the differential diagnosis of CHP [21]. Lung inflammation arises from the interplay between innate and adaptive immunity and alveolar epithelial cells, which include various subsets of intrapulmonary lymphocytes. Adaptive immunity is predominantly governed by conventional T cells, distinguished by their diverse T cell antigen receptor (TCR) repertoire. In contrast, innate immunity is mediated by innate lymphoid cell subsets, including innate-like T cells and innate lymphoid cells (ILCs), which lack TCRs [22]. Innate-like T cells, characterized by their expression of semi-invariant TCRs with a restricted repertoire diversity, include subsets such as γδT cells and mucosal-associated invariant T (MAIT) cells [23,24]. However, limited research has examined lymphocyte subsets, including ILCs, in BAL fluid, and the characteristics of BAL lymphocyte subsets and their roles in different ILDs remain poorly understood. This study aims to analyze lymphocyte subsets in BAL fluid from five different ILD groups, i.e., idiopathic interstitial pneumonia (IIP) (mainly NSIP), CHP, CTD, SAR, and CEP, to evaluate their potential diagnostic roles.

## 2. Materials and Methods

### 2.1. Study Participants

This prospective, non-interventional, observational study enrolled patients with ILD who underwent BAL between January 2016 and December 2019. Patients aged 20 years or older were recruited from the Juntendo University Hospital in Tokyo, Japan, and the Saitama Medical University Hospital in Saitama, Japan. ILD diagnoses were based on HRCT findings and elevated serum markers such as serum Krebs von den Lungen-6 (KL-6) and surfactant protein-D (SP-D). Patients were excluded if they met any of the following criteria: (1) diagnosis of an infectious disease, congestive heart failure, or cancer, (2) IPF cases identified by HRCT findings that did not necessitate bronchoscopy, or (3) cases considered inappropriate by the study investigators. We defined patients with IIP, including NSIP and COP (excluding IPF), based on the international IIP classification by the American Thoracic Society, the European Respiratory Society, and a previous report [6,25]. The diagnosis of CTD was established through a combination of medical history, physical examination, and specific autoantibody tests and subsequently confirmed by rheumatologists. Pulmonary SAR was histologically diagnosed in accordance with the clinical practice guidelines for SAR [26]. CEP was diagnosed based on the criteria established by Mochizuki et al. [27]. These include cases meeting at least two of the following three conditions: eosinophilia in BAL fluids of 10% or greater, eosinophilia in peripheral blood of 6% or greater, and eosinophil infiltration in transbronchial lung biopsy specimens. This study was conducted in compliance with the Declaration of Helsinki, approved by the Juntendo University Research Ethics Committee (Tokyo, Japan), and registered under approval numbers 15-171 (approved on 29 January 2016) and 16-030 (approved on 17 June 2016). Written informed consent was obtained from each patient before participation.

In addition to clinical parameters, we collected and analyzed peripheral blood leukocyte differential counts, serum markers, and differential counts of leukocytes and lymphocytes in BAL fluids.

### 2.2. Quantification of Lymphocyte Frequency in BAL Fluid and Peripheral Blood

Flow cytometry was performed as previously described [28]. Briefly, peripheral venous blood samples were collected in heparin-containing tubes, and peripheral blood mononuclear cells (3 × 10^6^/well) were purified by density-gradient centrifugation using the Ficoll–Paque Plus solution (Cytiva, Tokyo, Japan). The obtained BAL fluid was filtered through a 70 µm filter and centrifuged to obtain cells. The cells were stained with combinations of appropriate antibodies for 30 min at 4 °C. The following surface marker antibodies were used in this study: anti-CD3-APC-H7, anti-CD4-FITC, anti-CD19-FITC, anti-CD56-Alexa Fluor 700, anti-CD117 (c-Kit)-PE-CF594 (BD Biosciences, San Jose, CA, USA), anti-BDCA2-FITC, anti-CD1a-FITC, anti-CD11c-FITC, anti-CD14-FITC, anti-CD25-PE, anti-CD34-FITC, anti-CD123-FITC, anti-CD127 (IL-7Rα)-Brilliant Violet (BV) 421, anti-CD127-BV605, anti-CD161-PerCPCy5.5, anti-CD183 (CXCR3)-APC, anti-CD194 (CCR4)-BV510, anti-CD196 (CCR6)-PerCPCy5.5, anti-CD294 (CRTH2)-BV421, anti-CD69-APC, anti-CD69-Alexa Flour 700, anti-FCɛR1-FITC, anti-TCR-γδ-APC, and anti-Vα7.2-PE (BioLegend, San Diego, CA, USA). The lineage-negative (Lin^−^) markers defined were CD1a^−^, CD3^−^, CD11c^−^, CD14^−^, CD19^−^, CD34^−^, TCRγδ^−^, CD123^−^, BDCA2^−^, and FCɛR1^−^. The CD4^+^ Th1 cells identified were CD3^+^, CD4^+^, CCR4^−^, CCR6^−^, and CXCR3^+^ cells; the Th2 cells were CD3^+^, CD4^+^, CCR4^+^, CCR6^−^, and CXCR3^−^ cells; the Th17 cells were CD3^+^, CD4^+^, CCR4^+^, CCR6^+^, and CXCR3^−^ cells; the Treg cells were CD3^+^, CD4^+^, CD25^+^, and CD127^−^ cells; the γδT cells were CD3^+^ and TCRγδ^+^ cells; the MAIT cells were CD3^+^, Vα7.2, TCR^+^, and CD161high cells; the NK cells were CD3^−^ and CD56^+^ cells; the ILC1s were Lin^−^, CD127^+^, CD161^+^, CD117^−^, and CRTH2^−^ cells; the ILC2s were Lin^−^, CD127^+^, CD161^+^, and CRTH2^+^ cells; and the ILC3s were Lin^−^, CD127^+^, CD161^+^, CD117^+^, and CRTH2^−^ cells. After overnight fixation, the cells were analyzed using fluorescence-activated cell sorting (FACS) (LSRFortessa cell analyzer, BD Biosciences). The flow cytometry gating strategies are shown in the Appendix A. FACS data were analyzed using the FlowJo software (Version 9, BD Biosciences).

### 2.3. Statistical Analysis

Statistical significance between populations was determined using appropriate tests, including Fisher’s exact test. Comparisons across multiple groups were conducted using the Kruskal–Wallis test with Dunn’s multiple comparisons. Receiver operating characteristic (ROC) curve analysis was used to distinguish one ILD from another. Correlations between variables were evaluated using Spearman’s rank correlation coefficient. Statistical significance was set at *p* < 0.05. All statistical analyses were performed using the JMP Pro 18 software (JMP Statistical Discovery, Cary, NC, USA).

## 3. Results

### 3.1. Characteristics of Each ILD

Seventy-three patients with ILD who clinically required BAL were enrolled, of whom 56 patients diagnosed with CEP, SAR, CHP, CTD, and IIP (excluding IPF) were included in the analysis, whereas ILD subtypes, such as IPF and drug-induced ILD, were excluded. Among the 22 patients with IIP, eight were diagnosed with NSIP, two with COP, and 12 with unclassifiable IIP. Patients with CTD included those with rheumatoid arthritis (n = 4), anti-synthetase syndrome (n = 3), systemic sclerosis (n = 2), polymyositis (n = 1), dermatomyositis (n = 1), Sjögren syndrome (n = 1), and mixed connective tissue disease (n = 1). Additionally, five patients were unable to undergo the planned BAL procedure because of the worsening of their condition following enrollment. One patient experienced an adverse event following BAL, characterized by transient worsening of dyspnea which resolved spontaneously within 3 days without requiring steroid therapy. The characteristics of the 56 patients categorized by ILD subtype are presented in Table 1. The median ages and body mass index (BMI) of all patients were 72.0 years (interquartile range: 64.3–77.0) and 23.7 (interquartile range: 21.1–26.4), respectively. No significant differences were observed in these parameters among the ILD subtypes (Table 1). Of the total patients, 29 (51.8%) were male, 17 (77.3%) were in the IIP group, and only one (7.7%) was in the CTD group. Significant sex differences were observed among the ILD subtypes (*p* < 0.001) (Table 1). Before undergoing BAL, a total of four patients (7.4%) were on systemic steroids: two patients (15.4%) in the CTD group were taking 5 mg of prednisolone (PSL), one patient (16.7%) in the CEP group was taking 2.5 mg of PSL, and one patient (11.1%) in the CHP group was administered 10 mg of PSL (Table 1). Immunosuppressants were used by two patients (15.4%) in the CTD group (Table 1). Additionally, antifibrotic agents were administered to three patients (13.6%) in the IIP group and to two patients (22.2%) in the CHP group (Table 1). The median FEV_1_%, calculated as forced expiratory volume in 1 s (FEV_1_)/forced vital capacity, in the CEP group was 74.4% (interquartile range: 55.4–76.9), which was significantly lower than that in the other groups (*p* = 0.01) (Table 1). Serum KL-6 and SP-D levels, established biomarkers for assessing ILDs and indicative of epithelial damage in the lungs, were significantly higher in the CHP group than in the other groups, whereas KL-6 levels were lowest in the SAR group (*p* < 0.05) (Table 1). Serum C-reactive protein (CRP) levels, peripheral white blood cell (WBC) counts, and eosinophil counts were significantly higher in the CEP group, whereas the frequency of peripheral lymphocytes was significantly lower compared with the other groups (*p* < 0.05) (Table 1). In BAL cells, the CEP group exhibited significantly higher numbers and frequencies of eosinophils, as well as increased frequencies of ILC2 and ILC3, whereas the frequency of macrophages was significantly lower (*p* < 0.05) (Table 1, Figure 1 and Figure 2). The SAR group exhibited a significantly higher frequency of CD4^+^ T cells in the BAL fluid and an elevated CD4/CD8 ratio than the other groups (*p* < 0.001) (Table 1 and Figure 1). Furthermore, the frequency of Th1 cells in BAL fluids was significantly higher in the IIP group and significantly lower in the CTD group (*p* = 0.03) (Table 1 and Figure 2).

### 3.2. Parameters for Predicting the Diagnosis of Each ILD Subtype

ROC curve analysis was used to identify the optimal cutoff values for each parameter to differentiate between specific ILD subtypes and other ILDs. Table 2 and Table 3 present the optimal cutoff values for diagnosing specific ILD subtypes determined using JMP. The sensitivities and specificities are listed in the Appendix A. Parameters with an area under the ROC curve of 0.7 or higher that successfully distinguished CEP from other ILD subtypes included serum KL-6 (AUC 0.85; *p* = 0.047; cutoff value ≤ 577 U/mL), CRP levels (AUC 0.90; *p* = 0.005; cutoff value ≥ 2.2 mg/dL), peripheral blood WBC counts (AUC 0.85; *p* = 0.007; cutoff value ≥ 10,400 cells/mL), lymphocyte frequencies (AUC 0.94; *p* = 0.01; cutoff value ≤ 17.9%), neutrophil frequencies (AUC 0.71; *p* = 0.02; cutoff value ≤ 25.0%), BAL macrophage frequencies (AUC 0.81; *p* = 0.02; cutoff value ≤ 51.0%), BAL eosinophil frequencies (AUC 0.98; *p* = 0.009; cutoff value ≥ 8.0%) and counts (AUC 0.95; *p* = 0.01; cutoff value ≥ 5250 cells/mL), and BAL ILC3 frequencies (AUC 0.90; *p* = 0.04; cutoff value ≥ 0.0176%) (Table 2). Because CEP is strongly associated with increased peripheral blood eosinophil levels, the prediction of CEP occurrence by peripheral blood eosinophil values was unstable on logistic regression analysis. For distinguishing SAR from other ILD subtypes, the parameters with an area under the ROC curve of ≥ 0.7 included CD4^+^ T cells ≥ 65.2% (AUC 0.84; *p* = 0.03), CD8^+^ T cells ≤ 17.7% (AUC 0.90; *p* = 0.02), a CD4/CD8 ratio ≥ 3.95 (AUC 0.90; *p* = 0.009), and ILC1 frequencies ≥ 0.254% (AUC 0.73; *p* = 0.04) in BAL fluids (Table 2). In contrast, for CTD, the cutoff parameters in BAL fluids were CD4^+^ T cells ≤ 40.1% (AUC 0.77; *p* = 0.006), CD8^+^ T cells ≥ 29.4% (AUC 0.83; *p* = 0.002), and a CD4/CD8 ratio ≤ 1.2 (AUC 0.81; *p* = 0.03) (Table 2). Furthermore, parameters that differentiate IIP from other ILD subtypes included a BAL Th1 cell frequency of ≥ 17.4% (AUC 0.74; *p* = 0.009) (Table 2). For CHP, the distinguishing factors included serum KL-6 levels ≥ 2081 U/mL (AUC 0.70; *p* = 0.04) and serum SP-D levels ≥ 261 ng/mL (AUC 0.87; *p* = 0.009) (Table 2). For CTD, distinguishing factors included serum BAL neutrophil frequencies ≥ 2.0% (AUC 0.71; *p* = 0.02) and counts ≥ 4200 cells/mL (AUC 0.72; *p* = 0.03) (Table 2).

As there were insufficient parameters in the BAL fluid and peripheral blood that contributed to the diagnosis of IIP, CHP, and CTD, we subsequently used ROC curve analysis to determine the optimal cutoff values for each peripheral blood lymphocyte subset to differentiate these subtypes. The peripheral blood lymphocyte subsets that effectively distinguished IIP from CHP and CTD included a Th1 cell frequency ≥ 5.46% (AUC 0.73, *p* = 0.02) (Table 3). For CHP, the discriminative subsets had a Treg cell frequency ≥ 3.27% (AUC 0.73, *p* = 0.03) (Table 3). In contrast, peripheral blood lymphocyte subsets distinguishing CTD from IIP and CHP included a Th1 cell frequency ≤ 5.69% (AUC 0.86, *p* = 0.003), a Th17 cell frequency ≤ 1.62% (AUC 0.85, *p* = 0.003), and a Treg cell frequency ≤ 1.27% (AUC 0.78, *p* = 0.02) (Table 3).

Figure 3 demonstrates how the proposed cutoff values can be utilized to classify the five ILD subtypes. Initially, CEP is identified, followed by the classification of SAR based on BAL CD4 and CD8 levels. The figure further highlights the potential to distinguish IIP using BAL Th1 cells and CHP using serum KL-6 and SP-D levels. Additionally, after excluding CEP and SAR, it illustrates the possibility of differentiating IIP, CHP, and CTD based on Th1, Th17, and Treg cell frequencies in the peripheral blood lymphocyte fractions.

### 3.3. Correlation Between BAL WBC Subsets and Various Parameters

We examined whether the BAL cell subsets identified in Table 2 and Table 3 could be used to distinguish ILD subtypes correlated with respiratory function and peripheral blood parameters. BAL lymphocyte frequencies were positively correlated with DLco (*p* = 0.01) and serum KL-6 levels (*p* = 0.008), and negatively correlated with eosinophil frequency (*p* = 0.03) (Table 4). BAL lymphocyte counts were also positively correlated with KL-6 levels (*p* = 0.03), and neutrophil counts were positively correlated with CRP levels (*p* = 0.02) (Table 4). BAL eosinophils demonstrated positive correlations with CRP levels (*p* < 0.001), WBC counts (*p* < 0.001), neutrophil counts (*p* = 0.04), eosinophil counts (*p* < 0.001), and eosinophil frequency (*p* < 0.001), and negatively correlated with FEV_1_% (*p* = 0.02) and lymphocyte frequencies (*p* = 0.003) (Table 4). Similarly to eosinophils, ILC2 frequencies in BAL fluids were correlated with FEV_1_ (*p* = 0.04) (Table 4). Peripheral blood eosinophil frequencies also demonstrated positive correlations with the frequencies of lymphocytes (*p* = 0.03), Th2 cells (*p* = 0.03), ILC2s (*p* = 0.01), and ILC3s (*p* = 0.002) in BAL fluids (Table 4). Peripheral blood eosinophil counts were positively correlated with ILC2 (*p* = 0.03) and ILC3 (*p* = 0.02) frequencies in BAL fluids (Table 4). Additionally, BAL Th17 cell and ILC1 frequencies were negatively correlated with serum LD levels (*p* = 0.02 and 0.03, respectively), Th2 and Th17 cells, ILC1, and ILC3 frequencies were negatively correlated with KL-6 levels (*p* = 0.005, 0.003, 0.04, and 0.02, respectively), and ILC1 and ILC3 frequencies were negatively correlated with SP-D levels (*p* = 0.02) (Table 4). Furthermore, CRP showed a positive correlation with BAL CD8^+^ T cell (*p* = 0.048) and ILC3 frequencies (*p* = 0.03) and a negative correlation with CD4^+^ T cell frequencies (*p* = 0.04) (Table 4). The frequency of Th1 cells in BAL fluids was positively correlated with age (*p* = 0.04) and peripheral lymphocyte frequencies (*p* = 0.02) and negatively correlated with peripheral WBC counts (*p* = 0.04) (Table 4).

Furthermore, Table 5 presents the correlation between lymphocyte subsets in BAL fluids and peripheral blood after excluding CEP and SAR. NK cells in the peripheral blood were correlated with the total lymphocyte fraction in BAL fluids (*p* = 0.04) (Table 5). Moreover, significant positive correlations between BAL fluid and peripheral blood were observed only for Th1 cells (*p* = 0.047), Th17 cells (*p* = 0.01), and ILC1s (*p* = 0.04) (Table 5). Additionally, the frequency of Th1 cells in peripheral blood was positively correlated with CD4^+^ T cells (*p* = 0.009) and the CD4/CD8 ratio (*p* = 0.01), and negatively correlated with CD8^+^ T cells (*p* = 0.01) and ILC2 frequencies (*p* = 0.03) in BAL fluids (Table 5). BAL Th2 cells positively correlated with peripheral blood ILC1s (*p* = 0.04), BAL Th17 cells positively correlated with peripheral blood Th2 cells (*p* = 0.01), BAL Tregs positively correlated with peripheral blood Th17 cells (*p* = 0.045), BAL ILC1s exhibited a positive correlation with peripheral blood ILC1s (*p* = 0.04) and ILC3s (*p* = 0.02), and BAL ILC3s showed a positive correlation with peripheral blood ILC2s (*p* = 0.01) (Table 5).

## 4. Discussion

In this study, CEP exhibited a higher frequency of ILC2s and ILC3s, along with eosinophils, than the other four ILD subtypes. IIP showed an increased presence of Th1 cells, CHP demonstrated higher levels of Th17 and Treg cells, and CTD displayed lower frequencies of Th1, Th17, and Treg cells. To our knowledge, this is the first study to conduct a detailed analysis of lymphocyte subsets in BAL fluids from patients with ILD using flow cytometry. Additionally, it is the first study to propose that lymphocyte subset data may aid in the diagnosis of specific ILD subtypes.

This study highlights that CEP with eosinophilia in BAL fluid can be distinguished from the other four ILDs by several parameters, including BAL eosinophilia, elevated BAL ILC3 frequency, low serum KL-6 levels, high CRP levels, and elevated peripheral blood WBC counts. Infections need to be ruled out; however, the findings suggest that CEP exhibits a stronger inflammatory response than the other four ILD subtypes without an accompanying increase in KL-6. Th2 cells and ILC2s are well-known effector cells of eosinophilic inflammation [29,30]; however, this study identified an elevated proportion of ILC3s alongside ILC2s in BAL samples from patients with CEP, suggesting that increased BAL ILC3s may serve as a distinguishing feature of CEP from other ILDs. Despite the known association between ILC3 and obesity-related asthma, the BMI was not elevated in patients [31,32,33]. The observed correlation between peripheral blood eosinophils and not only ILC2s but also ILC3s in BAL fluid suggests that CEP is associated with type 2 inflammation and ILC3s activity. ILC3s are recognized as IL-17-producing cells, and lymphocytes such as Th17 cells that produce IL-17 have been associated with asthma severity. However, the role of IL-17-producing cells, including ILC3s, in CEP, whose etiology remains unclear, has yet to be elucidated, warranting further investigation. Furthermore, although the diagnostic criteria for CEP require a BAL eosinophil threshold of 25% or higher [11,34], this study suggests that CEP could be distinguished from the other four ILDs with a cutoff as low as 8% eosinophils in BAL fluids. Although BAL eosinophilia is known to occur in some ILDs aside from CEP [11,34], the absence of BAL eosinophilia in the four ILD subtypes, except CEP, included in this study, likely facilitated differentiation.

Similarly to CEP, SAR has distinctive features in the BAL fluid, allowing its classification to be separated from other ILD subtypes. Specifically, SAR is characterized by a high BAL CD4^+^ T-cell count, a low CD8^+^ T-cell count, and an elevated CD4/CD8 ratio, all of which are well-known markers of the condition [11,21]. In this study, a BAL CD4/CD8 ratio ≥ 3.95 was identified as the cutoff value for distinguishing SAR from other ILDs. This finding is consistent with those of previous reports, which suggest a ratio of 3.5 or higher, indicating SAR [11,35]. In contrast, IIP, CHP, and CTD often demonstrate a higher frequency of CD8^+^ T cells [11,21]; however, this marker was only useful for classifying CTD in this study. This study is the first to present data on CD4^+^ and CD8^+^ T cells in BAL fluids from patients with CTD, filling a gap with previously unavailable information [21]. Additionally, the high frequency of ILC1 in BAL fluids has emerged as a potential diagnostic marker for SAR, and the high frequency of BAL Th1 cells is indicative of IIP. The findings of this study regarding the significance of ILC1 in SAR align with previous reports suggesting that increased ILC1 accumulation in the blood is specific to patients with SAR compared to other granulomatous diseases and highlighting the potential of ILC1 as both a diagnostic biomarker and a therapeutic target for SAR [36]. CHP can be differentiated from the other subtypes based on significantly elevated serum KL-6 and SP-D levels. However, unlike CEP and SAR, which exhibit distinct markers, IIP, CHP, and CTD lack specific markers. Additional analyses, excluding CEPs and SARs, revealed that IIP could be classified based on the elevated frequency of peripheral blood Th1 cells, CHP by the increased frequency of peripheral blood Tregs, and CTD by the lower frequency of Th1, Th17, and Treg cells. As Th1, Th17, and Treg cells in the peripheral blood are all CD4^+^ helper T cells, this is consistent with the fact that a low frequency of CD4^+^ T cells in BAL fluids is a diagnostic aid for CTD.

The correlation between lymphocyte accumulation in BAL fluid and both impaired diffusion capacity and elevated serum KL-6 levels suggests a link between lymphocyte proliferation and alveolar damage. The neutrophil counts in BAL, a marker of inflammatory responses, correlated with serum CRP levels. Additionally, serum CRP levels were associated with high BAL CD8^+^ T cell and ILC3 levels, along with low CD4^+^ T cell levels, suggesting that CRP levels may be lower in SAR patients with a high BAL CD4/CD8 ratio and higher in CTD patients with a low BAL CD4/CD8 ratio. Eosinophil accumulation in BAL fluid correlated with serum CRP, elevated WBC, neutrophil, and eosinophil counts, as well as airflow limitation. Furthermore, the frequency of ILC2 in BAL fluid was positively associated with FEV_1_. These findings may be specific to CEP patients with asthma. Peripheral blood eosinophil levels were positively correlated with lymphocytes, Th2 cells, ILC2, and ILC3 in BAL fluid, suggesting that peripheral blood eosinophils reflect lung-specific type 2 inflammation. Furthermore, the negative correlation of BAL Th17 cells, ILC1, and ILC3 with serum LD, KL-6, and SP-D levels suggests that their cellular counterparts, i.e., type 2 inflammation, may be involved in alveolar epithelial damage. In CEP, which is driven by type 2 inflammation, serum KL-6 levels were lower, while BAL ILC3 levels were elevated. Moreover, after excluding CEP and SAR cases, the frequency of peripheral blood Th1 cells was associated with BAL CD4^+^ T cell levels and the CD4/CD8 ratio, further supporting the notion that low peripheral blood Th1 cell levels serve as a biomarker for CTD. The observed correlation between BAL WBC subsets and clinically established parameters reinforces previously recognized phenomena and corroborates this study’s findings.

This study had some limitations. First, it is a two-center, single-arm, open-label, observational study with a small sample size. Second, the comparison of only five ILD subtypes, the limited number of participants in each of the five ILD subtypes, the high proportion of patients with unclassifiable IIP within the IIP group, and the inability to clearly differentiate between fibrotic and cellular NSIP in cases classified as NSIP. In patients with IIP, CHP, and CTD (excluding IPF), BAL lymphocytes are potential indicators of the effectiveness of steroid treatments [11,21]. However, no significant differences in the BAL lymphocyte levels were observed among the subtypes. Furthermore, although peripheral blood parameters, such as KL-6 and NK cells, showed a positive correlation with BAL lymphocyte levels, BAL lymphocytes, KL-6, and NK cells did not distinguish between the three ILD subtypes. One possible explanation for this finding is the exclusion of IPF, which is typically associated with low BAL lymphocyte counts—an additional limitation of this study. This study identified several lymphocyte subsets in BAL fluids and peripheral blood that could potentially distinguish among ILD subtypes. In current clinical practice, CD4^+^ and CD8^+^ T cells in BAL fluids are routinely analyzed using flow cytometry. Therefore, even if incorporating ILC analysis presents certain challenges, implementing the analysis of CD4^+^ helper T-cell subsets would be relatively straightforward for integration into clinical practice. Furthermore, the ability to differentiate ILD subtypes using clinical information, including CT imaging, and biomarkers from BAL fluid and peripheral blood could eliminate the need for highly invasive biopsies in patients with ILD. This approach would also facilitate the differential diagnosis of severe cases where lung biopsies are challenging to perform. However, further studies are required to validate these findings and confirm their utility as diagnostic biomarkers.

## 5. Conclusions

This study underscores the potential utility of analyzing lymphocyte subsets in BAL fluids and peripheral blood as an adjunct to conventional diagnostic methods for distinguishing between various ILD subtypes. By identifying specific lymphocyte profiles, including BAL eosinophil and ILC3 frequencies in CEP, BAL CD4/CD8 ratios in SAR, and distinct peripheral blood Th1, Th17, and Treg cell patterns in IIP, CHP, and CTD, we found that these markers could aid in the accurate classification and differential diagnosis of ILDs. These findings contribute to a deeper understanding of the immunological mechanisms involved in ILD etiology and offer a promising approach for improving clinical diagnosis and treatment strategies. When combined with detailed clinical data and advanced thoracic imaging, such as HRCT, the analysis of BAL fluid and peripheral blood cell subsets may provide valuable insights for the diagnostic evaluation of ILD subtypes. Further validation using larger multicenter studies is required to confirm the clinical applicability of these markers in routine practice.

## Figures and Tables

**Figure 1 biomolecules-15-00122-f001:**
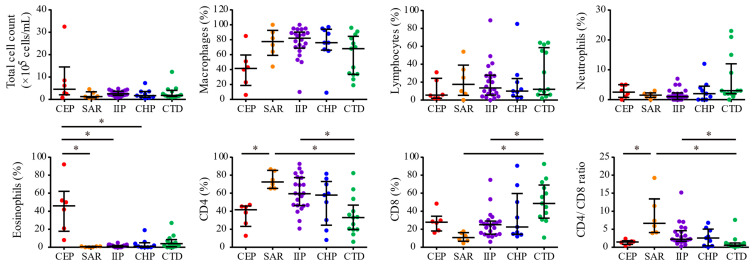
Total cell counts and percentages of cell fractions in BAL fluids. The data represent the median ± interquartile range. * *p* < 0.05; Dunn’s multiple comparisons.

**Figure 2 biomolecules-15-00122-f002:**
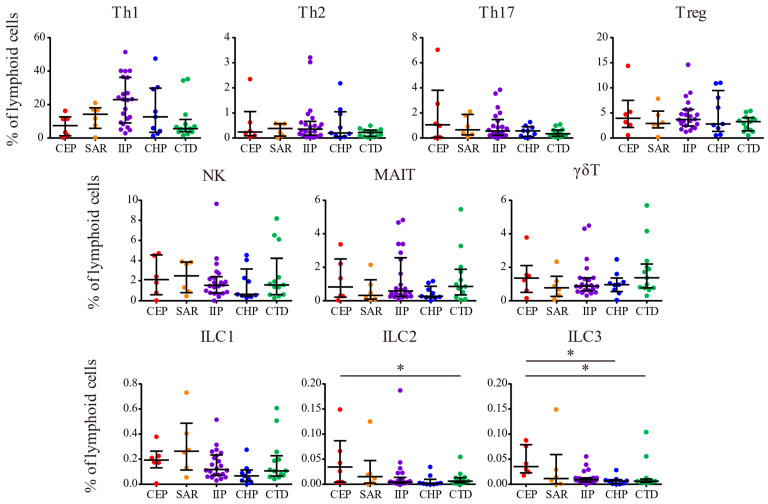
Frequencies of lymphocyte fractions in BAL fluids. The data represent the median ± interquartile range. * *p* < 0.05; Dunn’s multiple comparisons.

**Figure 3 biomolecules-15-00122-f003:**
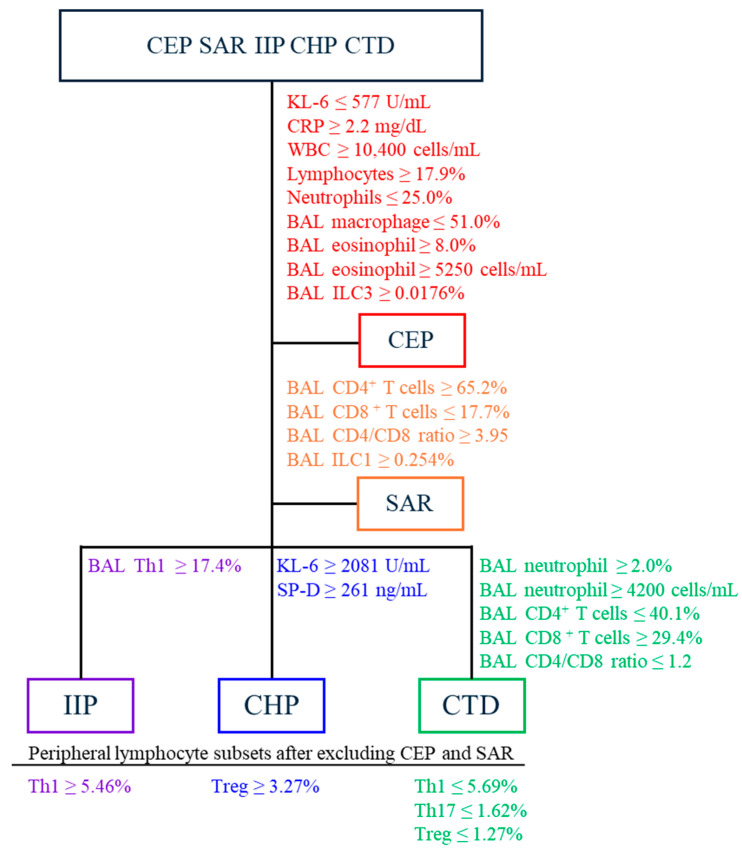
Utilization of the proposed cutoff values for the diagnosis of ILD subtypes.

**Table 1 biomolecules-15-00122-t001:** Characteristics of the study population.

	CEP	SAR	IIP	CHP	CTD	*p* Value
	n = 6	n = 6	n = 22	n = 9	n = 13	
Age (years)	66.0 (57.5–75.8)	54.5 (51.8–70.8)	74.5 (67.8–77.3)	70.0 (67.0–76.0)	72.0 (61.0–77.0)	0.17
Male, n (%)	2 (33.3)	4 (66.7)	17 (77.3)	5 (55.6)	1 (7.7)	<0.001 *
BMI (kg/m^2^)	20.6 (17.7–23.9)	23.3 (21.8–27.5)	25.5 (23.4–26.9)	23.0 (21.2–26.6)	21.4 (19.8–24.7)	0.045 *
Former smoker, n (%)	3 (50.0)	3 (50.0)	16 (72.7)	6 (66.7)	3 (23.1)	0.06
Bronchial asthma, n (%)	2 (33.3)	0 (0.0)	0 (0.0)	0 (0.0)	0 (0.0)	0.02 *
Allergic disease, n (%)	4 (66.6)	0 (0.0)	4 (18.2)	0 (0.0)	0 (0.0)	0.003 *
Systemic steroid use, n (%)	1 (16.7)	0 (0.0)	0 (0.0)	1 (11.1)	2 (15.4)	0.18
Immunosuppressant use, n (%)	0 (0.0)	0 (0.0)	0 (0.0)	0 (0.0)	2 (15.4)	0.28
Antifibrotic agent use, n (%)	0 (0.0)	0 (0.0)	3 (13.6)	2 (22.2)	0 (0.0)	0.34
VC (% predicted), n = 50	85.8 (71.3–99.7)	77.4 (59.6–110.8)	77.9 (63.2–86.9)	81.7 (58.9–86.8)	73.0 (59.9–90.6)	0.65
FVC (% predicted), n = 50	83.3 (72.5–101.7)	75.6 (61.0–106.7)	78.1 (60.1–87.3)	83.5 (60.2–86.9)	70.9 (58.1–87.5)	0.61
FEV_1_ (% predicted), n = 50	80.1 (59.1–96.4)	71.6 (56.1–100.1)	78.1 (66.4–85.2)	80.6 (63.5–93.7)	75.0 (65.6–93.5)	0.95
FEV_1_%, n = 50	74.4 (55.4–76.9)	79.7 (77.2–80.3)	83.2 (77.7–87.1)	85.9 (78.5–88.3)	85.4 (78.1–93.1)	0.01 *
DLco (% predicted), n = 45	44.7 (33.9–74.8)	56.6 (49.1–84.5)	34.5 (24.5–40.6)	34.2 (30.2–42.2)	42.3 (29.9–63.5)	0.10
DLco/V_A_ (% predicted), n = 44	59.5 (44.4–79.4)	92.9 (71.8–117.7)	66.0 (54.0–76.2)	67.6 (48.7–84.6)	71.2 (58.9–83.5)	0.24
Serum LD (U/L)	242 (186–355)	187 (166–231)	234 (213–251)	245 (214–269)	258 (217–300)	0.16
Serum KL–6 (U/mL), n = 53	307 (253–787)	200 (163–1461)	1155 (679–1766)	2081 (788–4126)	1344 (775–2049)	0.01 *
Serum SP–D (ng/mL), n = 43	161 (70.6–213)	NA	215 (144–354)	458 (266–784)	171 (132–232)	0.007 *^a^
Serum CRP (mg/dL)	3.53 (1.74–6.39)	0.10 (0.10–0.24)	0.30 (0.10–0.53)	0.40 (0.15–1.0)	0.25 (0.10–1.34)	0.009 *
Peripheral WBC (cells/μL)	10,905 (9025–16,610)	6800 (5095–8000)	5500 (4975–7150)	6300 (5650–8450)	6500 (5745–7300)	0.048 *
Peripheral lymphocytes (%)	13.6 (8.6–15.3)	25.8 (15.7–33.6)	28.8 (23.2–33.2)	27.4 (19.1–34.4)	28.5 (19.0–33.8)	0.01 *
Peripheral lymphocytes (cells/μL)	1442 (1204–1587)	1500 (1157–2167)	1574 (1366–1970)	1690 (1372–2061)	1700 (1174–2113)	0.66
Peripheral neutrophils (%)	55.9 (24.9–63.2)	59.5 (52.1–74.3)	62.9 (56.4–67.4)	62.9 (55.9–70.9)	61.1 (53.9–71.4)	0.59
Peripheral neutrophils (cells/μL)	6119 (2690–7556)	3869 (2646–5418)	3487 (2638–4917)	3755 (3461–5990)	3972 (3039–5155)	0.53
Peripheral eosinophils (%)	25.8 (16.8–57.0)	3.2 (2.0–6.3)	3.3 (1.5–3.9)	4.0 (2.8–6.0)	4.0 (1.7–6.8)	0.001 *
Peripheral eosinophils (cells/μL)	2973 (1417–8281)	160 (124–447)	176 (93.8–279.3)	244 (226–391)	240 (121–470)	<0.001 *
BAL fluid recovery (%)	50.5 (44.3–61.2)	71.4 (57.8–74.8)	55.0 (46.6–70.8)	64.0 (57.3–68.3)	61.3 (48.3–76.3)	0.22
BAL total cell count (×10^5^ cells/mL)	4.55 (2.08–14.5)	1.30 (0.75–3.40)	2.50 (1.80–3.68)	1.70 (0.95–3.55)	1.80 (1.35–4.05)	0.26
BAL macrophages (%)	41.5 (18.8–59.5)	77.5 (59.0–92.5)	82.0 (68.8–90.3)	76.0 (66.0–94.0)	68.0 (33.5–84.5)	0.04 *
BAL macrophages (×10^5^ cells/mL)	1.45 (0.98–2.31)	1.23 (0.60–1.73)	1.99 (1.41–2.63)	1.16 (0.65–1.95)	1.19 (0.83–1.84)	0.24
BAL lymphocytes (%)	5.5 (2.0–24.3)	17.5 (5.3–39.0)	13.5 (5.5–27.8)	10.0 (4.0–24.0)	12.0 (6.0–58.5)	0.56
BAL lymphocytes (×10^5^ cells/mL)	0.58 (0.05–0.92)	0.12 (0.06–1.58)	0.30 (0.12–0.73)	0.22 (0.04–0.67)	0.17 (0.09–1.80)	0.88
BAL neutrophils (%)	2.5 (0.75–5.0)	1.5 (0.75–2.25)	1.0 (0.0–2.25)	2.0 (0.0–4.5)	3.0 (2.0–12.0)	0.13
BAL neutrophils (×10^5^ cells/mL)	0.11 (0.01–0.16)	0.02 (0.01–0.04)	0.04 (0.00–0.05)	0.04 (0.00–0.13)	0.06 (0.04–0.25)	0.06
BAL eosinophils (%)	46.0 (17.8–62.0)	0.5 (0.0–1.0)	1.0 (0.0–2.0)	1.0 (0.0–5.0)	4.0 (0.5–8.5)	<0.001 *
BAL eosinophils (×10^5^ cells/mL)	2.16 (0.41–10.8)	0.00 (0.00–0.03)	0.03 (0.00–0.06)	0.01 (0.00–0.25)	0.06 (0.01–0.13)	0.002 *
BAL CD4 (%)	41.7(23.2–45.9)	72.5 (65.3–85.4)	59.4 (46.4–77.3)	57.9 (24.5–73.0)	33.0 (19.7–46.9)	<0.001 *
BAL CD8 (%)	27.9 (18.2–34.6)	10.8 (6.68–16.3)	25.1 (14.6–29.1)	22.5(14.5–59.6)	48.7 (32.4–69.0)	<0.001 *
BAL CD4/CD8 ratio	1.47 (0.82–1.83)	6.66 (4.09–13.4)	2.24 (1.65–4.61)	2.57 (0.41–5.06)	0.57 (0.28–1.19)	<0.001 *
BAL Th1 cells (% of lymphocytes)	7.41 (1.32–12.7)	14.2 (5.8–18.1)	23.0 (9.0–36.3)	12.7 (3.4–29.9)	5.6 (3.8–11.1)	0.03 *
BAL Th2 cells (% of lymphocytes)	0.24 (0.09–1.05)	0.38 (0.08–0.57)	0.36 (0.12–0.66)	0.20 (0.07–1.05)	0.22 (0.07–0.32)	0.58
BAL Th17 cells (% of lymphocytes)	1.06 (0.05–3.81)	0.66 (0.25–1.89)	0.57 (0.22–1.48)	0.57 (0.12–0.91)	0.34 (0.14–0.65)	0.48
BAL Treg cells (% of lymphocytes)	3.9 (2.1–7.5)	2.9 (2.0–5.4)	3.7 (2.4–5.7)	2.8 (1.3–9.5)	3.3 (1.4–4.1)	0.70
BAL NK cells (% of lymphocytes)	2.11 (0.60–4.55)	2.49 (0.81–3.87)	1.55 (0.78–2.41)	0.64 (0.46–3.17)	1.58 (0.61–4.23)	0.88
BAL MAIT cells (% of lymphocytes)	0.83 (0.22–2.50)	0.32 (0.10–1.26)	0.59 (0.27–2.57)	0.27 (0.19–0.87)	0.86 (0.35–1.89)	0.32
BAL γδT cells (% of lymphocytes)	1.36 (0.50–2.11)	0.78 (0.26–1.47)	0.88 (0.60–1.37)	0.96 (0.55–1.36)	1.38 (0.77–2.20)	0.58
BAL ILC1 cells (% of lymphocytes)	0.19 (0.13–0.26)	0.26 (0.11–0.49)	0.12 (0.07–0.24)	0.07 (0.02–0.11)	0.11 (0.07–0.23)	0.09
BAL ILC2 cells (% of lymphocytes)	0.03 (0.00–0.09)	0.02 (0.00–0.05)	0.00 (0.00–0.01)	0.00 (0.00–0.01)	0.01 (0.00–0.01)	0.046 *
BAL ILC3 cells (% of lymphocytes)	0.04 (0.02–0.08)	0.01 (0.00–0.06)	0.01 (0.00–0.01)	0.01 (0.00–0.01)	0.01 (0.00–0.01)	0.01 *
Peripheral Th1 cells (% of lymphocytes)	NA	7.8 (4.7–9.0)	7.8 (6.1–10.2)	6.9 (5.9–9.8)	4.4 (2.7–5.6)	0.003 *^b^
Peripheral Th2 cells (% of lymphocytes)	NA	3.0 (1.6–4.0)	2.0 (1.1–2.9)	2.0 (1.9–2.6)	1.9 (1.2–2.8)	0.61 ^b^
Peripheral Th17 cells (% of lymphocytes)	NA	1.7 (1.0–3.0)	1.8 (1.1–2.1)	2.3 (1.3–2.5)	0.99 (0.54–1.2)	0.001 *^b^
Peripheral Treg cells (% of lymphocytes)	NA	1.7 (0.92–2.5)	2.0 (1.6–2.8)	3.3 (1.7–4.7)	1.0 (0.75–2.1)	0.01 *^b^
Peripheral NK cells (% of lymphocytes)	NA	13.4 (3.8–25.5)	17.4 (12.4–24.5)	17.9 (11.7–24.3)	17.1 (11.3–32.0)	0.84 ^b^
Peripheral MAIT cells (% of lymphocytes)	NA	0.23 (0.14–0.45)	0.32 (0.15–0.69)	0.30 (0.11–0.61)	0.13 (0.08–1.3)	0.81 ^b^
Peripheral γδT cells (% of lymphocytes)	NA	4.7 (0.85–7.0)	2.7 (1.4–5.0)	3.1 (1.1–3.8)	2.9 (1.6–4.7)	0.96 ^b^
Peripheral ILC1 cells (% of lymphocytes)	NA	0.04 (0.03–0.05)	0.03 (0.02–0.05)	0.02 (0.01–0.12)	0.04 (0.01–0.06)	0.90 ^b^
Peripheral ILC2 cells (% of lymphocytes)	NA	0.01 (0.01–0.02)	0.01 (0.00–0.02)	0.01 (0.01 –0.01)	0.01 (0.01–0.05)	0.76 ^b^
Peripheral ILC3 cells (% of lymphocytes)	NA	0.00 (0.00–0.02)	0.01 (0.00–0.01)	0.00 (0.00–0.01)	0.00 (0.00–0.01)	0.52 ^b^

Data are presented as the median (interquartile range) or number (%). Allergic diseases included bronchial asthma (n = 2), allergic rhinitis (n = 1), and hay fever (n = 1) in the CEP group, but only hay fever in the IIP group. * *p* < 0.05: Fisher’s exact test or Kruskal-Wallis test; ^a^: analysis excluding SAR; ^b^: analysis excluding CEP. Abbreviations for all tables: AUC, area under the receiver operating characteristic curve; BAL, bronchoalveolar lavage; BMI, body mass index; CEP, chronic eosinophilic pneumonia; CHP, chronic hypersensitivity pneumonitis; CTD, connective tissue disease-associated interstitial lung disease; CRP, C-reactive protein; DLco, carbon monoxide diffusing capacity; FEV_1_, forced expiratory volume in 1 s; FEV_1_%, forced expiratory volume in 1 s/forced vital capacity; FVC, forced vital capacity; IIP, idiopathic interstitial pneumonia; ILC, innate lymphoid cell; KL-6, Krebs von den Lungen-6; MAIT, mucosal associated invariant T; NA, not applicable; NK, natural killer; γδT, gamma delta T; ROC, receiver operating characteristic; SAR, sarcoidosis; SP-D, surfactant protein-D; Treg, regulatory T cell; VA, alveolar volume; VC, vital capacity, WBC, white blood cells.

**Table 2 biomolecules-15-00122-t002:** ROC curves for predicting the diagnosis of each ILD subtype.

	CEP vs. the Others			SAR vs. the Others			IIP vs. the Others			CHP vs. the Others			CTD vs. the Others		
	AUC	*p* Value	Cutoff Value	AUC	*p* Value	Cutoff Value	AUC	*p* Value	Cutoff Value	AUC	*p* Value	Cutoff Value	AUC	*p* Value	Cutoff Value
Serum LD (U/L)	0.55	0.66		0.78	0.07		0.54	0.26		0.43	1.00		0.63	0.07	
Serum KL-6 (U/mL), n = 53	0.85	0.047 *	≤577	0.79	0.21		0.48	0.51		0.70	0.04 *	≥2081	0.60	0.85	
Serum SP-D (ng/mL), n = 43	0.72	0.25		NA			0.51	0.51		0.87	0.009 *	≥261	0.65	0.12	
Serum CRP (mg/dL)	0.9	0.005 *	≥2.2	0.75	0.17		0.58	0.52		0.50	0.43		0.47	0.50	
Peripheral WBC (cells/μL)	0.85	0.007 *	≥10,400	0.49	0.77		0.67	0.07		0.46	0.94		0.50	0.36	
Peripheral lymphocytes (%)	0.94	0.01 *	≤17.9	0.52	0.91		0.64	0.07		0.52	0.80		0.55	0.65	
Peripheral lymphocytes (cells/μL)	0.68	0.19		0.52	0.91		0.52	0.57		0.58	0.55		0.49	0.91	
Peripheral neutrophils (%)	0.71	0.02 *	≤25.0	0.51	0.81		0.55	0.56		0.55	0.63		0.51	0.46	
Peripheral neutrophils (cells/μL)	0.68	0.046 *	≥5789	0.50	0.97		0.61	0.15		0.55	0.58		0.50	0.70	
Peripheral eosinophils (%)	unstable			0.57	0.49		0.68	0.07		0.46	0.47		0.52	0.39	
Peripheral eosinophils (cells/μL)	unstable			0.57	0.56		0.70	0.06		0.42	0.54		0.52	0.45	
BAL total cell count (cells/mL)	0.71	0.11		0.66	0.31		0.44	0.43		0.61	0.55		0.52	0.80	
BAL macrophages (%)	0.81	0.02 *	≤51.0	0.57	0.46		0.64	0.06		0.60	0.45		0.64	0.13	
BAL macrophages (×10^5^ cells/mL)	0.50	0.90		0.63	0.35		0.68	0.04 *	≥1.5	0.62	0.28		0.58	0.34	
BAL lymphocytes (%)	0.68	0.26		0.53	0.98		0.49	0.78		0.55	0.76		0.61	0.16	
BAL lymphocytes (cells/mL)	0.48	0.66		0.57	0.68		0.47	0.45		0.40	0.83		0.55	0.21	
BAL neutrophils (%)	0.43	0.80		0.58	0.39		0.65	0.11		0.50	0.93		0.71	0.02 *	≥2.0
BAL neutrophils (cells/mL)	0.63	0.52		0.68	0.19		0.63	0.05		0.47	0.50		0.72	0.03 *	≥4200
BAL eosinophils (%)	0.98	0.009 *	≥8.00	0.75	0.15		0.65	0.08		0.57	0.49		0.38	0.71	
BAL eosinophils (×10^5^ cells/mL)	0.95	0.01 *	≥0.525	0.72	0.19		0.63	0.14		0.54	0.59		0.43	0.69	
BAL CD4 (%)	0.74	0.10		0.84	0.03 *	≥65.2	0.67	0.03 *	≥39.5	0.51	0.87		0.77	0.006 *	≤40.1
BAL CD8 (%)	0.48	0.66		0.90	0.02 *	≤17.7	0.60	0.1		0.52	0.54		0.83	0.002 *	≥29.4
BAL CD4/CD8 ratio	0.65	0.19		0.90	0.009 *	≥3.95	0.64	0.61		0.50	0.72		0.81	0.03 *	≤1.2
BAL Th1 cells (% of lymphocytes)	0.72	0.14		0.53	0.51		0.74	0.009 *	≥17.4	0.49	0.84		0.68	0.10	
BAL Th2 cells (% of lymphocytes)	0.53	0.67		0.50	0.58		0.59	0.25		0.51	0.66		0.65	0.11	
BAL Th17 cells (% of lymphocytes)	0.61	0.06		0.57	0.91		0.57	0.59		0.57	0.34		0.64	0.10	
BAL Treg cells (% of lymphocytes)	0.55	0.45		0.57	0.57		0.58	0.51		0.52	0.43		0.61	0.13	
BAL NK cells (% of lymphocytes)	0.57	0.78		0.55	0.8		0.50	0.66		0.60	0.45		0.51	0.43	
BAL MAIT cells (% of lymphocytes)	0.53	0.81		0.65	0.37		0.57	0.32		0.67	0.15		0.59	0.44	
BAL γδT cells (% of lymphocytes)	0.55	0.78		0.63	0.35		0.54	0.74		0.54	0.39		0.63	0.12	
BAL ILC1 cells (% of lymphocytes)	0.61	0.70		0.73	0.04 *	≥0.254	0.50	0.56		0.75	0.06		0.50	0.74	
BAL ILC2 cells (% of lymphocytes)	0.78	0.06		0.60	0.43		0.53	0.77		0.74	0.31		0.47	0.41	
BAL ILC3 cells (% of lymphocytes)	0.9	0.04 *	≥0.0176	0.53	0.2		0.48	0.24		0.70	0.20		0.60	0.80	

* *p* < 0.05.

**Table 3 biomolecules-15-00122-t003:** ROC curves of peripheral lymphocyte subset differentials for predicting the diagnosis of CHP, CTD, and IIP subtypes.

	IIP vs the Others			CHP vs the Others			CTD vs the Others		
% of Lymphocytes	AUC	*p* Value	Cutoff Value	AUC	*p* Value	Cutoff Value	AUC	*p* Value	Cutoff Value
Th1 cells	0.73	0.02 *	≥5.46	0.6	0.51		0.86	0.003 *	≤5.69
Th2 cells	0.49	0.49		0.44	0.88		0.53	0.53	
Th17 cells	0.62	0.16		0.76	0.07		0.85	0.003 *	≤1.62
Treg cells	0.58	0.55		0.73	0.03 *	≥3.27	0.78	0.02 *	≤1.27
NK cells	0.49	0.98		0.51	0.56		0.52	0.62	
MAIT cells	0.43	0.45		0.53	0.54		0.43	0.20	
γδT cells	0.51	0.49		0.55	0.4		0.53	0.96	
ILC1 cells	0.48	0.3		0.5	0.26		0.47	0.89	
ILC2 cells	0.54	0.16		0.56	0.36		0.59	0.048 *	≥0.026
ILC3 cells	0.63	0.17		0.58	0.68		0.58	0.24	

* *p* < 0.05.

**Table 4 biomolecules-15-00122-t004:** Correlation between BAL fluid WBC subsets and various parameters.

	BAL Lymphocytes (%)		BAL Lymphocytes (×10^5^ cells/mL)		BAL Neutrophils (%)		BAL Neutrophils (×10^5^ cells/mL)		BAL Eosinophils (%)		BAL Eosinophils (×10^5^ cells/mL)		BAL CD4 (%)		BAL CD8 (%)	
n = 56	rs	*p* Value	rs	*p* Value	rs	*p* Value	rs	*p* Value	rs	*p* Value	rs	*p* Value	rs	*p* Value	rs	*p* Value
Age (years)	0.09	0.53	0.07	0.62	0.04	0.75	0.12	0.36	−0.15	0.26	−0.13	0.35	0.16	0.25	−0.07	0.62
BMI (kg/m^2^)	0.21	0.13	0.26	0.06	0.01	0.92	0.10	0.46	−0.08	0.57	0.00	0.98	0.14	0.31	−0.03	0.83
VC (% predicted), n = 50	0.18	0.21	0.08	0.57	0.00	0.99	−0.06	0.68	0.08	0.59	0.00	0.98	0.08	0.59	−0.12	0.40
FVC (% predicted), n = 50	0.18	0.20	0.07	0.62	−0.03	0.84	−0.10	0.47	0.06	0.70	0.00	0.97	0.10	0.49	−0.15	0.28
FEV_1_ (% predicted), n = 50	0.12	0.39	0.04	0.79	0.01	0.93	0.00	0.97	−0.01	0.97	−0.06	0.66	0.10	0.47	−0.06	0.68
FEV_1_%, n = 50	−0.24	0.10	−0.20	0.16	−0.03	0.81	−0.02	0.87	−0.32	0.02 *	−0.32	0.02 *	0.02	0.90	0.07	0.63
DLco (% predicted), n = 45	0.38	0.01 *	0.25	0.10	0.00	0.98	−0.07	0.64	0.17	0.28	0.10	0.49	0.09	0.56	0.02	0.88
DLco/ V_A_ (% predicted), n = 44	0.25	0.10	0.15	0.35	0.00	0.98	−0.04	0.78	−0.07	0.65	−0.06	0.71	0.22	0.15	−0.04	0.79
Serum LD (U/L)	0.11	0.42	0.25	0.06	0.16	0.25	0.24	0.08	0.07	0.63	0.08	0.56	−0.14	0.29	0.16	0.23
Serum KL-6 (U/mL), n = 53	0.36	0.008 *	0.30	0.03 *	0.12	0.38	0.18	0.20	0.00	0.97	0.04	0.76	0.17	0.23	−0.02	0.87
Serum SP-D (ng/mL), n = 43	0.08	0.61	0.09	0.59	−0.04	0.80	−0.01	0.96	−0.16	0.32	−0.07	0.67	−0.02	0.89	0.03	0.86
Serum CRP (mg/dL)	−0.19	0.15	0.00	1.00	0.15	0.27	0.31	0.02 *	0.50	<0.001 *	0.48	<0.001 *	−0.28	0.04 *	0.27	0.048 *
Peripheral WBC (cells/mL)	−0.09	0.53	0.07	0.63	0.02	0.89	0.10	0.48	0.44	<0.001 *	0.43	<0.001 *	−0.22	0.11	0.12	0.36
Peripheral lymphocytes (%)	0.06	0.69	−0.10	0.46	−0.07	0.60	−0.12	0.36	−0.39	0.003 *	−0.39	0.003 *	0.16	0.24	−0.08	0.57
Peripheral lymphocytes (cells/mL)	−0.02	0.87	−0.04	0.79	−0.07	0.63	−0.06	0.65	−0.03	0.83	−0.03	0.84	−0.05	0.74	0.05	0.70
Peripheral neutrophils (%)	0.10	0.45	0.02	0.88	0.05	0.72	0.00	1.00	−0.05	0.73	−0.04	0.76	0.05	0.69	0.01	0.93
Peripheral neutrophils (cells/mL)	−0.07	0.61	0.03	0.83	−0.04	0.75	0.00	0.97	0.28	0.04 *	0.28	0.04 *	−0.12	0.36	0.12	0.39
Peripheral eosinophils (%)	−0.29	0.03 *	−0.09	0.49	0.10	0.45	0.20	0.13	0.46	<0.001 *	0.45	<0.001 *	−0.22	0.10	0.12	0.39
Peripheral eosinophils (cells/mL)	−0.24	0.07	−0.03	0.85	0.11	0.41	0.24	0.07	0.53	<0.001 *	0.53	<0.001 *	−0.24	0.07	0.17	0.20
	**BAL CD4/CD8 Ratio**		**BAL Th1 Cells** **(% of Lymphocytes)**		**BAL Th2 Cells** **(% of Lymphocytes)**		**BAL Th17 Cells** **(% of Lymphocytes)**		**BAL Treg Cells** **(% of Lymphocytes)**		**BAL ILC1 cells (% of Lymphocytes)**		**BAL ILC2 Cells (% of Lymphocytes)**		**BAL ILC3 Cells (% of Lymphocytes)**	
**n = 56**	**rs**	***p*** **Value**	**rs**	***p*** **Value**	**rs**	***p*** **Value**	**rs**	***p*** **Value**	**rs**	***p*** **Value**	**rs**	***p*** **Value**	**rs**	***p*** **Value**	**rs**	***p*** **Value**
Age (years)	0.12	0.37	0.28	0.04 *	0.00	0.98	0.08	0.54	0.05	0.73	0.02	0.87	−0.13	0.33	−0.11	0.43
BMI (kg/m^2^)	0.08	0.57	0.09	0.49	−0.06	0.65	−0.11	0.43	−0.02	0.89	−0.24	0.08	0.10	0.47	−0.12	0.39
VC (% predicted), n = 50	0.09	0.54	0.14	0.35	−0.26	0.07	−0.20	0.15	−0.15	0.29	−0.07	0.64	−0.01	0.96	0.10	0.48
FVC (% predicted), n = 50	0.12	0.40	0.16	0.26	−0.26	0.06	−0.22	0.12	−0.13	0.37	−0.05	0.72	−0.02	0.88	0.08	0.57
FEV_1_ (% predicted), n = 50	0.09	0.54	0.14	0.33	−0.20	0.16	−0.24	0.09	−0.19	0.19	−0.18	0.21	−0.14	0.32	−0.01	0.97
FEV_1_%, n = 50	0.00	0.99	−0.15	0.30	0.05	0.73	−0.14	0.34	−0.14	0.34	−0.16	0.26	−0.29	0.04 *	−0.07	0.64
DLco (% predicted), n = 45	0.02	0.92	0.08	0.58	−0.15	0.34	−0.12	0.44	−0.19	0.22	0.08	0.61	0.13	0.41	−0.21	0.17
DLco/ V_A_ (% predicted), n = 44	0.12	0.45	0.14	0.36	0.06	0.68	0.00	0.98	−0.06	0.69	0.09	0.58	0.05	0.76	−0.35	0.02 *
Serum LD (U/L)	−0.17	0.20	−0.04	0.76	−0.22	0.10	−0.31	0.02 *	−0.15	0.28	−0.28	0.03 *	−0.09	0.52	−0.19	0.16
Serum KL-6 (U/mL), n = 53	0.10	0.48	0.26	0.06	−0.38	0.005 *	−0.40	0.003 *	−0.26	0.06	−0.28	0.04 *	−0.12	0.39	−0.31	0.02 *
Serum SP-D (ng/mL), n = 43	−0.02	0.90	0.12	0.46	−0.02	0.88	−0.19	0.23	−0.16	0.31	−0.34	0.02 *	−0.04	0.78	−0.35	0.02 *
Serum CRP (mg/dL)	−0.29	0.03 *	−0.19	0.16	0.16	0.23	0.11	0.44	0.09	0.53	−0.04	0.75	0.12	0.37	0.29	0.03 *
Peripheral WBC (cells/mL)	−0.18	0.19	−0.27	0.04 *	0.02	0.87	0.01	0.92	0.02	0.90	−0.17	0.21	0.09	0.51	0.07	0.63
Peripheral lymphocytes (%)	0.14	0.30	0.32	0.02 *	0.04	0.79	−0.01	0.92	0.05	0.70	0.01	0.96	−0.18	0.20	−0.32	0.02 *
Peripheral lymphocytes (cells/mL)	−0.03	0.81	0.05	0.71	0.03	0.84	−0.01	0.92	0.02	0.86	−0.18	0.18	−0.14	0.31	−0.35	0.008 *
Peripheral neutrophils (%)	0.02	0.86	−0.05	0.69	−0.08	0.57	0.00	1.00	−0.06	0.68	−0.02	0.88	−0.09	0.52	−0.01	0.94
Peripheral neutrophils (cells/mL)	−0.12	0.37	−0.22	0.11	0.02	0.89	0.07	0.59	0.02	0.90	−0.14	0.29	0.01	0.93	0.03	0.82
Peripheral eosinophils (%)	−0.19	0.17	−0.19	0.16	0.28	0.03 *	0.22	0.10	0.11	0.42	0.10	0.49	0.34	0.01 *	0.41	0.002 *
Peripheral eosinophils (cells/mL)	−0.22	0.10	−0.26	0.05	0.25	0.06	0.18	0.18	0.09	0.53	−0.05	0.74	0.30	0.03 *	0.32	0.02 *

* *p* < 0.05.

**Table 5 biomolecules-15-00122-t005:** Correlation between BAL lymphocyte and peripheral lymphocyte subsets in CHP, CTD, and IIP subtypes.

n = 44	BAL Lymphocytes (%)		BAL Lymphocytes (×10^5^ cells/mL)		BAL CD4 (%)		BAL CD8 (%)		BAL CD4/CD8 Ratio		BAL Th1 Cells (% of Lymphocytes)	
% of Lymphocytes	rs	*p* Value	rs	*p* Value	rs	*p* Value	rs	*p* Value	rs	*p* Value	rs	*p* Value
Peripheral Th1 cells	−0.14	0.36	−0.15	0.32	0.39	0.009 *	−0.37	0.01 *	0.38	0.01 *	0.30	0.047 *
Peripheral Th2 cells	−0.06	0.71	0.02	0.88	−0.09	0.58	0.07	0.67	−0.07	0.67	−0.09	0.57
Peripheral Th17 cells	−0.14	0.35	−0.12	0.43	0.14	0.37	−0.17	0.26	0.17	0.27	0.02	0.90
Peripheral Treg cells	−0.29	0.05	−0.29	0.05	0.01	0.94	−0.10	0.54	0.05	0.76	−0.17	0.27
Peripheral NK cells	0.32	0.04 *	0.34	0.02 *	0.10	0.51	−0.09	0.55	0.10	0.53	0.16	0.29
Peripheral MAIT cells	−0.06	0.69	−0.03	0.85	−0.04	0.80	0.04	0.80	−0.03	0.85	−0.05	0.74
Peripheral γδT cells	−0.08	0.63	−0.14	0.36	−0.01	0.96	0.02	0.90	−0.02	0.92	0.08	0.61
Peripheral ILC1 cells	−0.14	0.35	−0.18	0.25	0.12	0.43	−0.19	0.23	0.17	0.26	0.09	0.58
Peripheral ILC2 cells	−0.24	0.11	−0.23	0.14	0.07	0.67	−0.02	0.88	0.03	0.83	0.01	0.94
Peripheral ILC3 cells	−0.11	0.49	−0.13	0.40	0.20	0.19	−0.27	0.08	0.25	0.10	0.20	0.19
	**BAL Th2 Cells** **(% of Lymphocytes)**		**BAL Th17 Cells** **(% of Lymphocytes)**		**BAL Treg Cells** **(% of Lymphocytes)**		**BAL ILC1 Cells (% of Lymphocytes)**		**BAL ILC2 Cells (% of Lymphocytes)**		**BAL ILC3 Cells (% of Lymphocytes)**	
**% of Lymphocytes**	**rs**	***p* Value**	**rs**	***p* Value**	**rs**	***p* Value**	**rs**	***p* Value**	**rs**	***p* Value**	**rs**	***p* Value**
Peripheral Th1 cells	0.21	0.18	0.19	0.21	0.24	0.11	−0.08	0.63	−0.33	0.03 *	−0.07	0.67
Peripheral Th2 cells	0.28	0.07	0.38	0.01 *	0.18	0.24	−0.07	0.64	0.17	0.26	0.11	0.47
Peripheral Th17 cells	0.28	0.07	0.38	0.01 *	0.30	0.045 *	−0.07	0.64	−0.11	0.48	0.11	0.48
Peripheral Treg cells	0.18	0.25	0.19	0.21	0.16	0.31	−0.03	0.86	−0.19	0.21	0.24	0.12
Peripheral NK cells	−0.18	0.25	−0.23	0.13	−0.20	0.20	0.11	0.48	0.11	0.49	0.03	0.86
Peripheral MAIT cells	0.02	0.92	−0.03	0.83	0.00	0.99	−0.20	0.19	0.28	0.06	0.01	0.94
Peripheral γδT cells	0.14	0.37	0.05	0.74	0.12	0.45	0.03	0.86	−0.13	0.40	0.14	0.38
Peripheral ILC1 cells	0.31	0.04 *	0.20	0.18	0.06	0.70	0.31	0.04 *	0.07	0.64	0.16	0.29
Peripheral ILC2 cells	0.28	0.06	0.28	0.07	0.24	0.12	0.27	0.07	0.11	0.49	0.37	0.01 *
Peripheral ILC3 cells	0.22	0.16	0.16	0.31	0.05	0.74	0.36	0.02 *	0.07	0.65	0.23	0.13

* *p* < 0.05.

## Data Availability

Data are contained within the article and Appendix A.

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
