# Peer review of "Evaluating the Diagnostic Value of Lymphocyte Subsets in Bronchoalveolar Lavage Fluid and Peripheral Blood Across Various Diffuse Interstitial Lung Disease Subtypes"

_biomolecules, 2025, doi:10.3390/biom15010122_

Round 1

Reviewer 1 Report

Comments and Suggestions for Authors

In the paper: Evaluating the Diagnostic Value of Lymphocyte Subsets in Bronchoalveolar Lavage Fluid and Peripheral Blood Across Various Diffuse Interstitial Lung Disease Subtypes, the authors examined a broad spectrum of selected immune cells in bronchoalveolar lavage fluid and peripheral blood of 56 patients from five types of diffuse Interstitial Lung Diseases.

They found an increase of Eosinophils and ILC3 in BALF in Chronic Eosinophilic Pneumonia. In Sarcoidosis CD4/CD8 ratios in BALF and type 1 innate lymphoid cell were identified as differentiation markers. BALF Th1 cells, neutrophils, and CD4/CD8 ratio, serum KL-6 and SP-D were detected as distinguishing markers for Intersticial Idiopathic Pneumonia, Chronic Hyoersensitivity Pneumonitis, and Connective Tissue Disease-associated interstitial pneumonia. The authors concluded that results of this analysis offer a new insight into inflammatory characteristics between various ILD which has diagnostic value. The study is comprehensive and brings new information about immune characteristics of diffuse ILD.

Minor revision

1. What is the frequency of diffuse ILD in the population, generally? should be mention in the paper. 

2. How the mechanisms of the pathogenesis in CEP and Sar are related to identified markers of differentiation for these diseases, the authors should discuss this.

3. What would be the benefits of using results of this analysis in diagnostics of diffuse ILD, authors should emphasize this in Discussion.

Reviewer 2 Report

Comments and Suggestions for Authors

Dear Authors,

Many thanks for your manuscript:

Evaluating the Diagnostic Value of Lymphocyte Subsets in Bronchoalveolar Lavage Fluid and Peripheral Blood Across Various Diffuse Interstitial Lung Disease Subtypes

[Manuscript _biomolecules_3398242]

Below you may find some comments, along with my overall impression and recommendations, that you might find useful for your manuscript.
